# A Deep Learning Approach to Predict Autism Spectrum Disorder Using Multisite Resting-State fMRI

Faria Zarin Subah [1], Kaushik Deb [1,*], Pranab Kumar Dhar [1] and Takeshi Koshiba [2]

1 Department of Computer Science and Engineering, Chittagong University of Engineering & Technology, Chattogram 4349, Bangladesh; u1504027@student.cuet.ac.bd (F.Z.S.); pranabdhar81@cuet.ac.bd (P.K.D.)

2 Faculty of Education and Integrated Arts and Sciences, Waseda University, 1-6-1 Nishiwaseda, Shinjuku-ku, Tokyo 169-8050, Japan; tkoshiba@waseda.jp

* Correspondence: debkaushik99@cuet.ac.bd

**Abstract:** Autism spectrum disorder (ASD) is a complex and degenerative neuro-developmental disorder. Most of the existing methods utilize functional magnetic resonance imaging (fMRI) to detect ASD with a very limited dataset which provides high accuracy but results in poor generalization. To overcome this limitation and to enhance the performance of the automated autism diagnosis model, in this paper, we propose an ASD detection model using functional connectivity features of resting-state fMRI data. Our proposed model utilizes two commonly used brain atlases, Craddock 200 (CC200) and Automated Anatomical Labelling (AAL), and two rarely used atlases Bootstrap Analysis of Stable Clusters (BASC) and Power. A deep neural network (DNN) classifier is used to perform the classification task. Simulation results indicate that the proposed model outperforms state-of-the-art methods in terms of accuracy. The mean accuracy of the proposed model was 88%, whereas the mean accuracy of the state-of-the-art methods ranged from 67% to 85%. The sensitivity, F1-score, and area under receiver operating characteristic curve (AUC) score of the proposed model were 90%, 87%, and 96%, respectively. Comparative analysis on various scoring strategies show the superiority of BASC atlas over other aforementioned atlases in classifying ASD and control.

**Keywords:** autism spectrum disorder; resting-state fMRI; predefined brain atlas; ABIDE; functional connectivity; connectivity matrix; deep neural network

## 1. Introduction

Autism spectrum disorder (ASD) is a range of lifelong neurodevelopmental disorders characterized by difficulties in social interaction and communication and by restricted and repetitive patterns of behavior [1]. According to an estimate conducted by the World Health Organization (WHO), 1 in 160 children suffers from ASD worldwide [2]. It is associated with an array of behavioral symptoms which may take a drastic form if the diagnosis is delayed [3,4]. Although symptoms are prevalent during infancy, diagnosis is delayed in most cases. It is because the current diagnostic procedure of ASD is purely subjective and interview-based that requires the physician to go through a child's developmental history and behavior [5,6]. Though these methods are quite accurate, they are undoubtedly exhaustive, extensive, and also require professional expertise that might not be available at many health institutions.

Recently, with the advancement of technology, a large cohort of studies are considering an automated computer-aided diagnosis of autism [7–9] and also developing interactive tools to aid in the rehabilitation and treatment of autistic patients [10–12]. Such automated approaches would decrease subjectivity and improve diagnostic reproducibility and availability. It would also play a substantial role in ensuring early diagnosis. Magnetic resonance imaging (MRI) can be used to detect various neuropsychiatric and neurodegenerative disorders, such as schizophrenia [13–16], dementia, depression [17], autism [18–21], ADHD [22], Alzheimer's [23,24], etc., by observing anatomical patterns of the brain using

structural MRI data or by connecting changes in the brains' functional architecture to psychiatric health conditions using functional MRI data.

## 1.1. Autism Detection Using Structural MRI Data

Structural MRI studies emphasize the morphometric and volumetric investigation to detect abnormal brain anatomy. Riddle et al. [25] observed that the grey matter volume, the left anterior superior temporal gyrus, and the total brain volume are enlarged in autistic children by 1–2% approximately after conducting a voxel-based morphometry analysis. These findings were not consistent at the adult stage. While Aylward et al. [26] measured total brain volumes and head circumference from 1.5 m coronal MRI scans in 67 autistic subjects and 83 healthy community volunteers within the age range of 8 to 46 years and concluded that no volumetric differences exist between ASD and control brains aged above 12. Palmen et al. [27] concluded that high-functioning autistic subjects showed an enlargement of grey-matter volume but no increase in white-matter and cerebellar volume. In this regard, Courchesne et al. [28] also reported increased grey matter volume, specifically in the temporal lobes in autism. On the contrary, an increased white matter and reduced cerebral cortex and hippocampus-amygdala were found in the autistic brain by Herbert et al. [29]. Conversely, Jou et al. [30] observed decreased central white matter volume in autistic subjects performing experiments using MRI data. Thus, all the previous works utilizing structural MRI data failed to reach strong conclusions regarding volumetric changes and presented inconsistent findings regarding grey and white matter volumes in autistic and control brains. The use of a small sample size and limited age range can be regarded as a limitation. In fact, these studies focused more on discovering common patterns among ASD versus control group rather than solving the inherent classification problem with reliable accuracy [31].

However, Kong et al. [32] presented a promising study to solve the classification problem using only structural MRI data by constructing an individual brain network for each subject and extracting connectivity features between each pair of ROIs (region of interests). Then, these features were ranked by Fisher score computation. The top 3000 features were provided as input to a deep neural network classifier. Results showed a very high accuracy of 90.39% and an area under receiver operating characteristic curve (AUC) score of 97.38% using only 182 subjects from a single site. Accuracy above 0.9 was obtained when only a dozen subjects were considered [33], and accuracy deteriorates consequently when a larger dataset is introduced [34].

## 1.2. Autism Detection Using Resting-State Functional MRI Data

Nielsen et al. [34] used whole-brain point-to-point functional connectivity, including resting-state functional magnetic resonance imaging (fMRI) data comprising 964 subjects collected from 16 different international sites. Raw image data were preprocessed in MATLAB (Mathworks, Natick, MA, USA) [35] using SPM8 (Wellcome Trust, London, UK) distributor software [36]. After preprocessing, mean BOLD signals for each subject were extracted from 7266 grey matter ROIs. A 7266 × 7266 association matrix representing functional connectivity between every ROI pair was computed from Pearson correlation coefficients for every subject. Each ROI pair was defined as a connection. Connections were categorized into several bins, and a general linear model classifier was fitted on bins containing connections. A very low accuracy of merely 60% was achieved for whole-brain classification.

Heinsfeld et al. [37] applied a deep learning algorithm combining a multilayer perceptron (MLP) along with autoencoders and obtained 70% mean classification accuracy using connectivity features derived from the Craddock 200 (CC200) brain atlas. However, it took a huge training time (over 32 h). Eslami et al. [38] proposed ASD-Diagnet, a framework for detecting ASD using only resting-state fMRI data. A combined learning procedure using an autoencoder and a single-layered perceptron (SLP) was used. Furthermore, to increase the number of training subjects, a data augmentation strategy based on linear interpolation

on available feature vectors was implemented. However, this strategy could only improve accuracy by 1% and presented a mean classification accuracy of 70.1%. Tang et al. [39] performed ASD and control classification utilizing two types of fMRI imaging modalities. A 3D Resnet and MLP classifier were used in the classification process. This approach obtained 74% classification accuracy.

To compete with the advancement of technologies in the neuroimaging field, there exists a scope for improving the accuracy of existing methods related to diagnosing autism spectrum disorder utilizing functional MRI. The primary goal of this study is to achieve a superior accuracy to the existing methods that have utilized mean time series signal based ASD detection and also to reduce the training time. In this regard, we have utilized the resting-state fMRI data of 866 subjects comprising 402 ASD and 464 control subjects from the Autism Brain Imaging Data Exchange (ABIDE) dataset. A detailed description of this dataset has been represented in Section 3.1, Table 1.

The principal contributions of this work are listed below:

1. Achieved maximum accuracy of 88% using only resting-state functional MRI (rs-fMRI) data.
2. Reduced training time to less than 10 min, whereas other studies required more than 32 h in [37] and 41 min in [38].
3. Reached the conclusion that the Bootstrap Analysis of Stable Clusters (BASC) atlas using 122 ROIs yield a higher predictive power than other predefined atlases from comparative analysis.

The subsequent contents of this study are organized as mentioned. Section 2 and its subsections describe our proposed model to detect ASD in detail. Section 3 describes the experimental results, discussions, and overall performance analysis for our proposed model. Section 4 contains a brief conclusion regarding our research findings and implications.

## 2. Proposed Approach

Resting-state fMRI (rs-fMRI) performs brain mapping to evaluate regional interactions occurring in a task-negative state [40]. Studies mostly rely on access to raw fMRI image data, but raw data take up a huge amount of processing time and might suffer from overfitting due to their high dimensionality. Taking into account the above issues, a deep learning approach to detect autism spectrum disorder from functional connectivity features derived from preprocessed fMRI data is proposed. Figure 1 illustrates each step of the proposed approach.

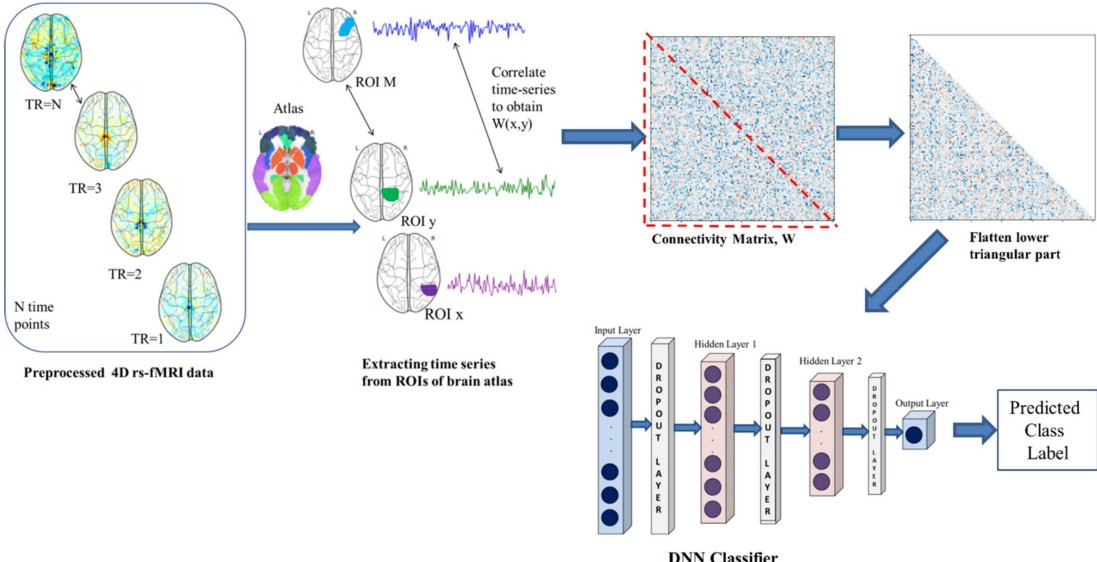

**Figure 1.** The architecture of the proposed approach for autism spectrum disorder (ASD) detection.

A step-by-step description of the proposed approach is represented in the following subsections.

## 2.1. Preprocessing

The preprocessed fMRI dataset of the ABIDE repository [41,42] was collected from the preprocessed connectome project (PCP) [43]. The configurable pipeline for the analysis of connectomes (CPAC) preprocessing pipeline was used, which included slice timing correction, motion correction, intensity normalization, nuisance signal removals, such as respiration, heartbeat, low-frequency scanner drifts, global mean signal regression, head motion, etc. The preprocessed data were band-pass filtered (0.01–0.1 Hz) and spatially registered to MNI152 template space. Detailed information regarding algorithms, strategies, parameters used, and software implemented can be obtained from [44].

## 2.2. Time-Series Extraction from ROIs Using Brain Atlas

Functional MRI scans produce a set of three-dimensional images recorded over time and measure a signal (most commonly, the blood-oxygen-level-dependent signal or BOLD signal) that is related to neural activity. In this case, a subject lies in the MRI scanner without thinking or doing anything in particular, while a series of brain images are generated over time that depicts the change in BOLD signal intensity [40]. Thus, a single preprocessed fMRI scan is a 4D time-series data including three spatial dimensions and time.

Instead of working with the entire time series obtained from every brain voxel directly, certain brain regions of interest (as defined by the brain atlases) were considered here. The mean time-series signals or BOLD signal intensities from voxels enclosed within those regions were extracted using brain atlases. Four standard predefined brain atlases were used to extract ROIs. Among these, the Bootstrap Analysis of Stable Clusters (BASC) atlas, the Power atlas and the Craddock 200 (CC200) atlas are functional and Automated Anatomical Labeling (AAL) is a structural atlas. The number of ROIs defined by the BASC, Power, CC200, and AAL atlas were 122, 264, 200, and 116, respectively. Information regarding these atlases is provided in Section 2.2.1.

### 2.2.1. Selection of Predefined Atlases

(i) AAL—Automated Anatomical Labelling: It is a structural atlas comprising 116 ROIs defined from the anatomy of a reference object [45]. These ROIs are represented in continuous colors in Figure 2 along the three anatomical planes (Axial, Sagittal, and Coronal).

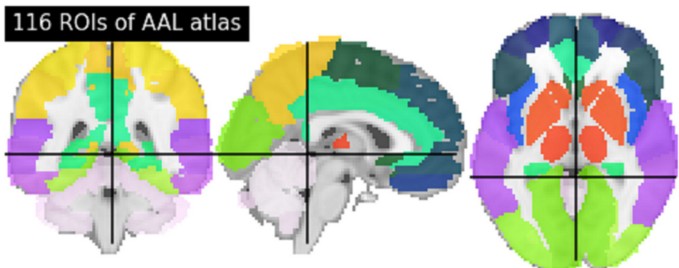

**Figure 2.** Automated Anatomical Labelling (AAL) atlas.

(ii) BASC—Bootstrap Analysis of Stable Clusters This multiscale functional brain parcellation atlas was generated from rs-fMRI images using a method called bootstrap analysis of stable clusters in [46]. It consists of a different number of ROIs {36, 64, 122, 197, 325, 444}. The BASC atlas with 122 ROIs was utilized in this study which is represented in Figure 3 using continuous colors.

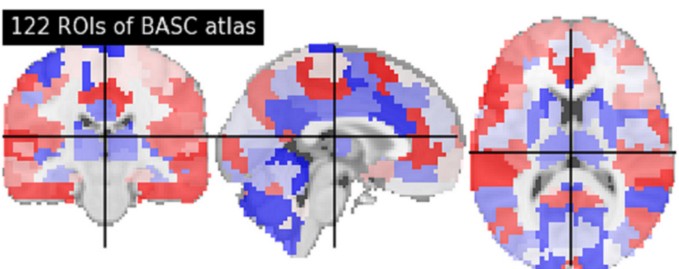

**Figure 3.** Bootstrap Analysis of Stable Clusters (BASC) atlas.

(iii)   CC200—Craddock 200 The CC200 functional brain parcellation atlas was generated by normalized cut spectral clustering of the entire brain into 200 spatially-constrained regions of homogeneous functional activity by Craddock et al. [47].

(iv)   Power Power atlas comprising 264 ROIs was defined by local graph-connectivity by Power et al. [48]. All the images used in this section were generated by the Nilearn Python library [49].

### 2.2.2. Mean Timeseries Extraction of ROIs from 4D fMRI Brain Volume

Applying a brain atlas on 4D fMRI scans can be imagined as overlaying a series of 3D grids that act like a mask and select which cubes or voxels to sample from at every time point. Thus, original 4D fMRI data of a single subject with dimensions (H, W, D, T) gets transformed to 2D data with dimensions (T, N) where H, W, D, T, and N represents height, width, depth, or number of slices, time points of the image volume and number of ROIs, respectively.

However, the whole process of extracting mean time-series signals applying brain atlas on preprocessed rs-fMRI data requires a huge amount of memory space. Due to hardware and memory constraints, we utilized pre-extracted time-series data. In this study, mean time-series signals containing CC200 and AAL defined ROIs were obtained directly from PCP [44], and those containing BASC and Power atlas defined ROIs were collected from [50]. Thus, the (T, N) dimension in our study was (196, 200), (196, 116), (196, 122), and (196, 264) for CC200, AAL, BASC, and Power, respectively, since scans were generated for 196 time points in each case.

### 2.3. Building Functional Connectivity Matrix

The (T, N) dimensional data were then transformed into a functional connectivity matrix or a connectome with dimensions (N, N). A functional connectome can be defined as a connectivity matrix that measures the correlation between a set of individual brain ROIs as defined by the respective brain atlas. In this case, dimensions became (200, 200), (116, 116), (122, 122), and (264, 264) for CC200, AAL, BASC, and Power atlases, respectively.

To build functional connectomes, tangent embedded parametrization of the default Ledoit-Wolf regularized covariance estimator was implemented using the Nilearn library [49]. Tangent space embedding uses both correlations and partial correlations to capture reproducible connectivity patterns at the group-level and models individual connectivities as deviations from the mean [51]. Functional connectivity matrices are represented in Figure 4 as an embedded connectome to visualize the striking differences between functional connectivity among brain regions from a random autistic and control sample.

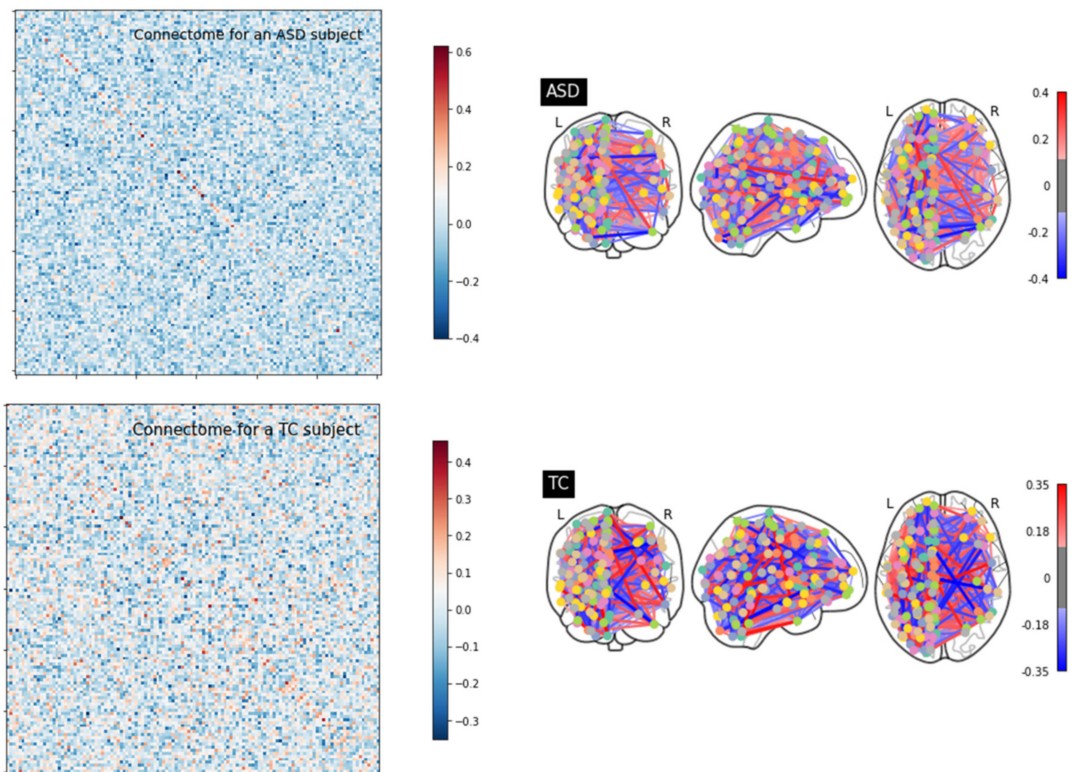

**Figure 4.** Difference between connectomes of random participants belonging to the ASD and control group.

### 2.4. Transforming 2D Functional Connectivity Matrix to 1D Feature Vector

The tangent connectivity matrix was symmetrical, and the upper triangular value repeated the lower one. To reduce dimensionality, the upper triangular values, including the principal diagonal, were removed, and the lower triangular values were retrieved as shown in Figure 1. Next, the lower triangular part was flattened to a 1D feature vector of size,

$$S = \frac{N(N-1)}{2},\tag{1}$$

where $N$ = number of ROIs. Thus, using the stated atlases, we received a feature vector of size 19,900, 6670, 7381, and 34,716, respectively, for CC200, AAL, BASC, and Power atlases in the case of each subject.

### 2.5. Classification Using a Deep Neural Network Classifier

The obtained feature vectors in Section 2.4 were provided as input to the proposed deep neural network classifier (DNN). The proposed DNN (referred as Model-2 in the later sections) consisted of two hidden layers with 32 neurons per layer, as illustrated in Figure 5. A dropout layer with a dropout probability of 0.8 was introduced between each layer to control overfitting. The hidden layers used the rectified linear unit (ReLU) activation function, and the final output layer used the sigmoid activation function.

Let $x_i$ and $b_i$ be the input and bias value of hidden layer $i$, respectively, $W_i$ is the weight vector connecting the nodes in hidden layer $i$ to the nodes in hidden layer $i + 1$, then, hidden layer $i + 1$ is activated using the following equation:

$$Z_{i+1} = f(W_i x_i + b_i)\tag{2}$$

where $Z$ denotes activation of the subscripted hidden layer and $f$ is the ReLU activation function defined as:

$$f(x) = \max(0, x)\tag{3}$$

Thus, ReLU gives an output within the range of [0, ∞]. In the case of sigmoid,

$$f(x) = \frac{1}{1 + e^{-x}} \tag{4}$$

where $e$ = Euler's number. It exists between 0 and 1 and predicts the probability value as an output in the case of binary classification. Xavier and He weight initializers were used with sigmoid and ReLU activations, respectively. Adam, with a relatively low learning rate of 0.0001 and default parameters, was applied as the optimizer [52]. The binary cross-entropy loss function was used in this binary classification problem (ASD vs. control). This loss function is defined as:

$$J = -\frac{1}{m} \sum_{i=1}^{m} [y_i \cdot \log(p(y_i)) + (1 - y_i) \cdot \tag{5}$$

where $m$ is the total number of samples, $y$ is the label, and $p$ indicates the probability of $y$ belonging to autism or control group. The objective of the network is to minimize the value of loss function, $J$.

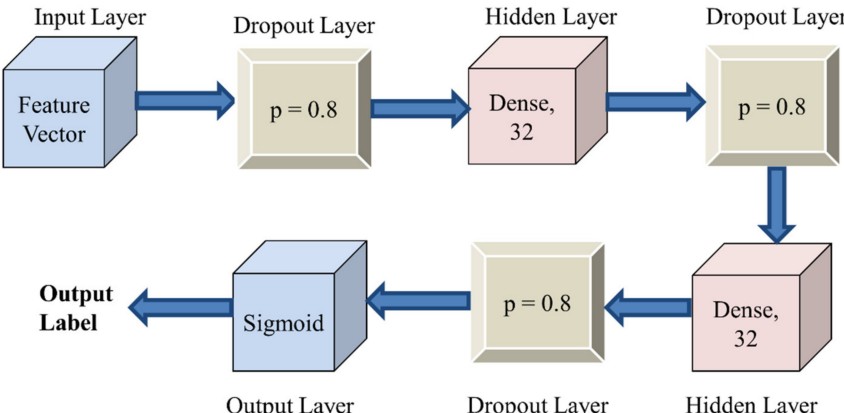

**Figure 5.** Proposed deep neural network architecture to predict ASD.

Since our input vector was high dimensional, we employed L2 regularization techniques and introduced a very small batch size of 10 during training so that our model generalized well and performed better on unseen data.

## 3. Experimental Results and Discussion

The experiments were implemented in Colab notebook using python v3.6. Cloud provided backend free tensor processing unit (TPU) runtime and 12.72 GB RAM on Intel Core i7 processor. The deep learning model used Keras API with TensorFlow backend. For model evaluation, scoring strategy, and numerical computation, the scikit learn library was utilized.

### 3.1. ABIDE Dataset Description

Experimental analysis was conducted over the ABIDE dataset. ABIDE is a consortium that has collected resting-state fMRI data and corresponding phenotypic information of subjects from 17 international sites [41]. Originally, it contained 1112 scans, including 539 ASD and 573 control individuals. However, all functional data could not pass the QAP (Quality Assessment Protocol) metrics as formulated by the PCP community [44], which reduced the size of the dataset to 866 subjects containing 402 ASD and 464 control subjects. Table 1 contains the phenotypic information of the participants used in this study.

**Table 1.** Phenotypic information summary of the participants from the ABIDE dataset.

| Site | Count | | | Age Range |
| --- | --- | --- | --- | --- |
| | **ASD** | **Control** | **Total** | |
| Caltech | 5 | 10 | 15 | 17–56 |
| CMU | 6 | 4 | 10 | 19–40 |
| KKI | 12 | 20 | 32 | 8–13 |
| LEUVEN | 26 | 30 | 56 | 12–32 |
| MAX_MUN | 19 | 27 | 46 | 7–58 |
| NYU | 74 | 98 | 172 | 6–39 |
| OHSU | 12 | 13 | 25 | 8–15 |
| OLIN | 14 | 14 | 28 | 10–24 |
| PITT | 24 | 26 | 50 | 9–35 |
| SBL | 12 | 14 | 26 | 20–64 |
| SDSU | 8 | 18 | 26 | 9–17 |
| Stanford | 12 | 13 | 25 | 8–13 |
| Trinity | 19 | 25 | 44 | 12–26 |
| UCLA | 48 | 37 | 85 | 8–18 |
| UM | 46 | 73 | 119 | 8–29 |
| USM | 43 | 24 | 67 | 9–50 |
| YALE | 22 | 18 | 40 | 8–18 |
| TOTAL | 402 | 464 | 866 | 6–64 |

*The table is under the heading:* **Autism Brain Imaging Data Exchange (ABIDE) Dataset**

Detailed information about ABIDE dataset is available in [42].

### 3.2. Data Partitioning Using Stratified 5-Fold Cross-Validation

Extensive fine-tuning and experimentation were performed in the DNN classifier by varying different hyperparameters. The number of hidden layer neurons of the deep neural network was varied within the range of 8 to 64, and performance was recorded in each case using each of the four atlases. The network structure represented in Figure 5 outperformed other configurations. Each network was validated using the stratified 5-fold cross-validation approach preserving the percentage of subjects in each target class (autism and control) to retain class balance. Twenty percent of the dataset was used as test cases, and the remaining 80% was utilized in training and validation. Within the training dataset, 80% of data was used for training and 20% for validation. A figurative representation of data partitioning is shown in Figure 6. This strategy allowed robust model evaluation while training and testing using different subsets of data.

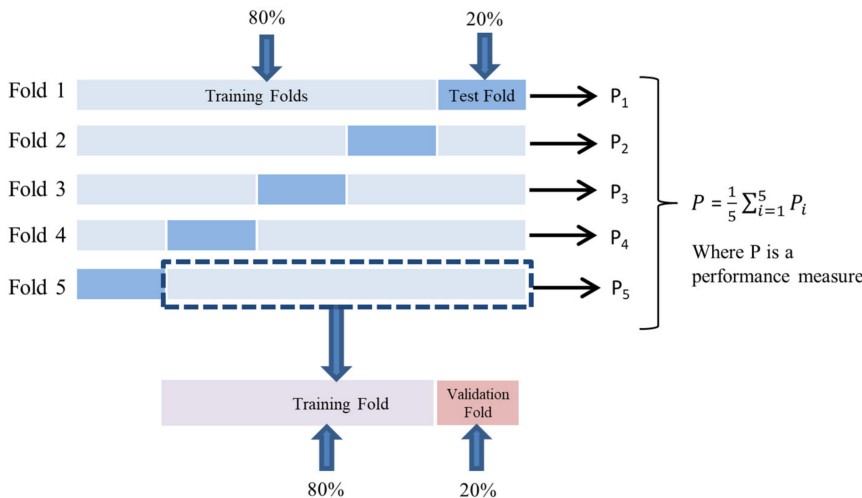

**Figure 6.** Data partitioning using stratified 5-fold cross-validation.

### 3.3. Performance Evaluation Using Different Atlas

Mean performance metrics for different network configurations by fine-tuning using the atlases mentioned in Section 2.2.1 are represented in the following subsections.

Tables 2–5 represent the value of sensitivity, F1-score, and AUC score for each atlas. Values of mean accuracy and its standard deviation are also shown in percentages. A dropout probability of 0.8 was introduced between each layer to control overfitting, as shown in Figure 5. All the models were compiled using the Adam optimizer with a learning rate of 0.0001 and binary cross-entropy loss function and trained with a batch size of 10. Confusion matrix and mean AUC (area under receiver operating characteristic curve) curves were also represented in each case.

**Table 2.** Mean performance evaluation using Craddock 200 (CC200) atlas for different network configurations.

| | Network Configuration | | | Mean Performance Evaluation using Craddock 200 (CC200) Atlas | | | | |
|---|---|---|---|---|---|---|---|---|
| Model | Input Layer | Hidden Layer 1 | Hidden Layer 2 | Accuracy | Acc. Std (%) | Sensitivity | F1-Score | AUC Score |
| Model-1 | 19900 | 64 | 32 | 0.8473 | 1.57 | 0.9406 | 0.8510 | 0.9515 |
| **Model-2** | **19900** | **32** | **32** | **0.8668** | **2.38** | **0.8683** | **0.8579** | **0.9571** |
| Model-3 | 19900 | 32 | 16 | 0.8530 | 3.02 | 0.7194 | 0.8185 | 0.9569 |
| Model-4 | 19900 | 16 | 16 | 0.6843 | 2.74 | 0.9429 | 0.7343 | 0.9595 |
| Model-5 | 19900 | 16 | 8 | 0.5947 | 4.23 | 0.9182 | 0.6770 | 0.9592 |

**Table 3.** Mean performance evaluation using Power atlas for different network configurations.

| | Network Configuration | | | Mean Performance Evaluation using Power Atlas | | | | |
|---|---|---|---|---|---|---|---|---|
| Model | Input Layer | Hidden Layer 1 | Hidden Layer 2 | Accuracy | Acc. Std (%) | Sensitivity | F1-Score | AUC Score |
| Model-1 | 34716 | 64 | 32 | 0.7898 | 2.12 | 0.9626 | 0.8098 | 0.9513 |
| **Model-2** | **34716** | **32** | **32** | **0.8533** | **2.38** | **0.8633** | **0.8453** | **0.9531** |
| Model-3 | 34716 | 32 | 16 | 0.8245 | 2.31 | 0.9429 | 0.8335 | 0.9505 |
| Model-4 | 34716 | 16 | 16 | 0.8638 | 3.22 | 0.7662 | 0.8385 | 0.9565 |
| Model-5 | 34716 | 16 | 8 | 0.5993 | 2.16 | 0.9802 | 0.6946 | 0.9509 |

**Table 4.** Mean performance evaluation using BASC atlas for different network configurations.

| | Network Configuration | | | Mean Performance Evaluation using Bootstrap Analysis of Stable Clusters (BASC) Atlas | | | | |
|---|---|---|---|---|---|---|---|---|
| Model | Input Layer | Hidden Layer 1 | Hidden Layer 2 | Accuracy | Acc. Std (%) | Sensitivity | F1-Score | AUC Score |
| Model-1 | 7381 | 64 | 32 | 0.8557 | 2.76 | 0.8634 | 0.8467 | 0.9570 |
| **Model-2** | **7381** | **32** | **32** | **0.8787** | **2.33** | **0.9029** | **0.8739** | **0.9587** |
| Model-3 | 7381 | 32 | 16 | 0.8672 | 2.49 | 0.8507 | 0.8563 | 0.9439 |
| Model-4 | 7381 | 16 | 16 | 0.8545 | 2.51 | 0.8358 | 0.8419 | 0.9471 |
| Model-5 | 7381 | 16 | 8 | 0.8579 | 1.90 | 0.8731 | 0.8511 | 0.9418 |

**Table 5.** Mean performance evaluation using AAL atlas for different network configurations.

| | Network Configuration | | | Mean Performance Evaluation using Automated Anatomical Labeling (AAL) Atlas | | | | |
|---|---|---|---|---|---|---|---|---|
| Model | Input Layer | Hidden Layer 1 | Hidden Layer 2 | Accuracy | Acc. Std (%) | Sensitivity | F1-Score | AUC Score |
| Model-1 | 6670 | 64 | 32 | 0.8611 | 2.59 | 0.8933 | 0.8561 | 0.9523 |
| **Model-2** | **6670** | **32** | **32** | **0.8737** | **2.49** | **0.8412** | **0.8599** | **0.9512** |
| Model-3 | 6670 | 32 | 16 | 0.8679 | 3.77 | 0.7941 | 0.8475 | 0.9500 |
| Model-4 | 6670 | 16 | 16 | 0.8702 | 3.95 | 0.9082 | 0.8665 | 0.9522 |
| Model-5 | 6670 | 16 | 8 | 0.8312 | 2.73 | 0.9404 | 0.8379 | 0.9509 |

### 3.3.1. CC200 Atlas

Analyzing the mean performance evaluation using five different network configurations from Table 2, it can be observed that CC200 achieved the highest accuracy and F1-score in our proposed Model-2. Though, sensitivity and AUC were the highest for Model-4. However, accuracy and F1 score were relatively lower. Model-2 achieved a relatively good score across all performance metrics. Figure 7 represents the confusion matrix and mean AUC curve using the proposed Model-2.

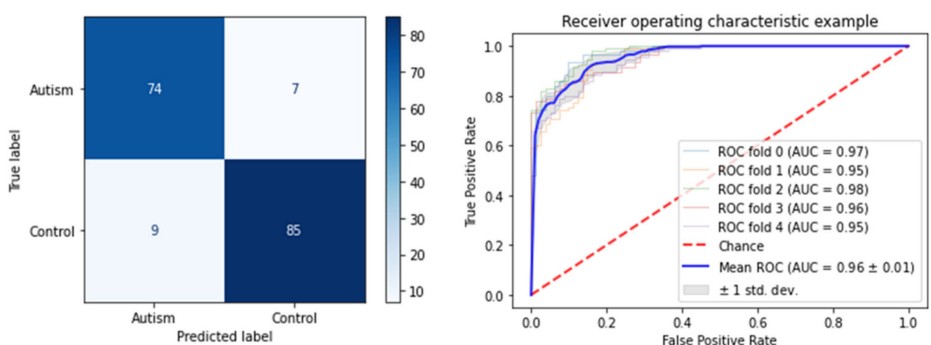

**Figure 7.** Confusion matrix and area under receiver operator characteristic curve (AUC) curve for CC200 atlas using Model-2.

### 3.3.2. Power Atlas

From Table 3, it is observed that the proposed Model-2 achieved a superior F1-score while the remaining scores were also greater than 85%. Model-5 obtained the highest sensitivity while accuracy and F1-score remained poor. AUC score remained almost constant at 95% for all models. Figure 8 represents the confusion matrix and mean AUC curve using the proposed Model-2.

### 3.3.3. BASC Atlas

Table 4 shows that the BASC atlas achieved the highest performance measure across all performance metrics using the proposed model. Figure 9 contains the confusion matrix and AUC curve using Model-2.

### 3.3.4. AAL Atlas

From Table 5, it is evident that AAL atlas represented a fluctuating performance across different models for each scoring metric. The confusion matrix and AUC curve using Model-2 are illustrated in Figure 10.

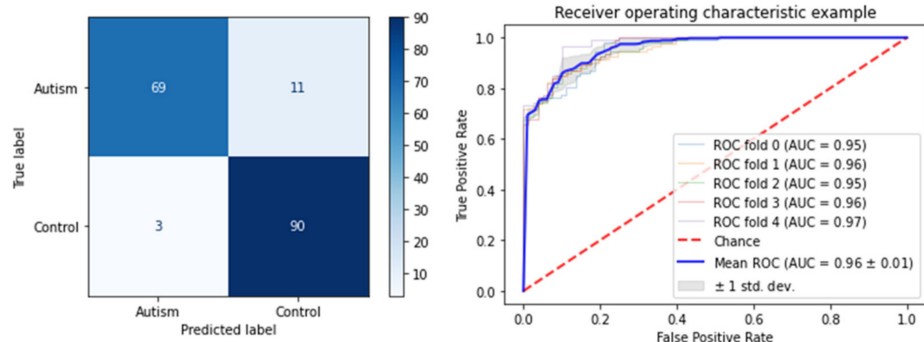

**Figure 8.** Confusion matrix and AUC curve for Power atlas using Model-2.

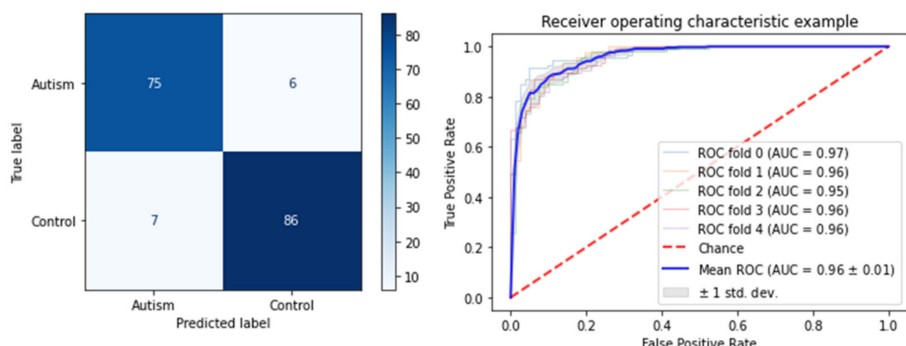

**Figure 9.** Confusion matrix and AUC curve for BASC atlas using Model-2.

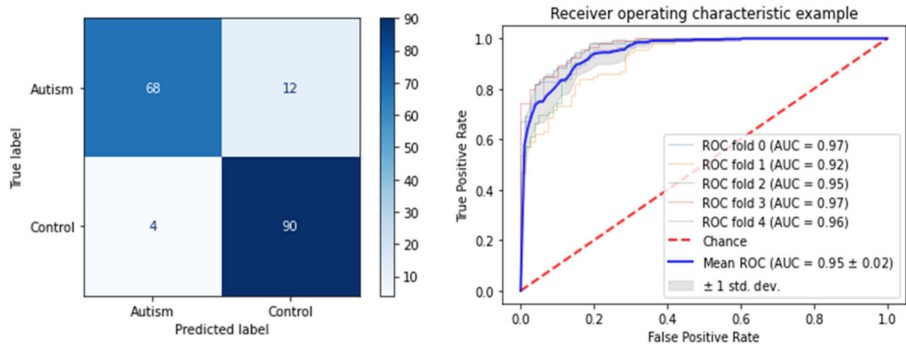

**Figure 10.** Confusion matrix and AUC curve for AAL atlas using Model-2.

### 3.4. Performance Comparison among Atlases

A comparison of all performance metrics across all four atlases is represented in Figure 11 using the proposed Model-2 to determine which atlas had the most discriminative power in identifying autism and control cases.

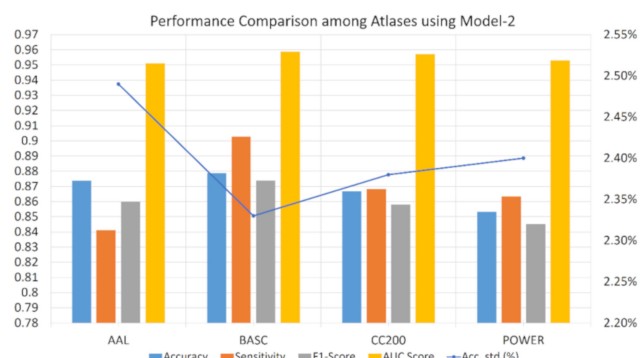

**Figure 11.** Performance comparison among atlases using Model-2.

From the above graphical analysis, the following points can be demonstrated:

- BASC atlas provided superior performance in terms of accuracy, sensitivity, F1, and AUC score using the proposed Model-2.
- AAL showed inconsistent results among various metrics. It had the lowest sensitivity value, which is very crucial and significant in medical diagnosis.
- CC200 and Power atlas depicted the lowest predictive power based on its performance value across all measures.

From the aforementioned points, it can be concluded that the BASC atlas displays the best performance across all metrics. It exhibits the highest discriminative power in a balanced manner which is evident from its F1 score. Other models exceeding two hidden layers were also attempted, but experimental results deteriorated due to the limited dataset.

### 3.5. Performance Comparison Using BASC Atlas and Single-Site Data

A quantitative analysis of accuracy, sensitivity, and F1-score obtained by testing the proposed Model-2 classifier on data obtained from individual screening sites of ABIDE is represented in Table 6 using BASC atlas.

**Table 6.** Performance comparison using Model-2 and BASC atlas on data obtained from individual screening sites.

| Site ID | No of Subjects | Accuracy | Sensitivity | F1-Score |
|---------|---------------|----------|-------------|----------|
| PITT | 50 | 0.94 | 0.96 | 0.94 |
| YALE | 40 | 0.95 | 0.91 | 0.95 |
| NYU | 172 | 0.92 | 0.92 | 0.91 |
| UM | 119 | 0.93 | 0.92 | 0.92 |

From the above tabular analysis, a significant improvement in performance can be observed across different performance metrics while using data obtained from individual screening sites. On the contrary, performance drops when the entire ABIDE dataset comprising 17 international sites was used for testing. This is because different sites use different MRI acquisition protocols, scanning parameters, ways of laying the participants in the scanner, etc., which introduces huge variance across datasets obtained from different sites. Moreover, the effect of domain shift and distributional shift might also be responsible for such differences in performance measures.

### 3.6. Performance Comparison with Machine Learning Methods

Performance of machine learning classifiers, such as k-nearest neighbors (KNN), Random Forest, Naïve Bayes, etc., to successfully predict functional connectivity-based classification have been compared across rs-fMRI cohorts by Dadi et al. in [50]. To our knowledge, no such comparative analysis has been conducted using deep learning classifiers as of now. In Table 7, a performance comparison between popular machine learning algorithms and our proposed Model-2 is represented. From this table, it is evident that our proposed deep learning model outperformed the machine learning techniques.

**Table 7.** Performance comparison with different machine learning algorithms.

| | L-SVM | KNN | DT | RF | GNB | Model-2 |
|-------|-------|------|------|------|------|---------|
| AAL | 0.6613 | 0.481 | 0.5224 | 0.5637 | 0.6176 | 0.8737 |
| BASC | 0.6166 | 0.5473 | 0.5115 | 0.5427 | 0.62 | 0.8787 |
| CC200 | 0.6865 | 0.5488 | 0.5166 | 0.574 | 0.6026 | 0.8668 |
| POWER | 0.6697 | 0.5265 | 0.5161 | 0.5254 | 0.6062 | 0.8533 |

Here, L-SVM indicates linear support vector machine; KNN means k-nearest neighbor; DT represents decision tree; RF indicates random forest and GNB means Gaussian Naïve Bayes classifier.

Performance was measured using accuracy metrics. The Green–Yellow–Red color scale is used to highlight the performances where green indicates superior performance and red indicates the lowest performance.

### 3.7. Performance Comparison with Existing Literature

Table 8 illustrates the highest performance measure obtained from different existing works related to fMRI based ASD identification using brain atlases and our proposed model.

**Table 8.** Performance comparison with existing literature.

| Methods | Year Published | Accuracy (%) |
|---|---|---|
| Abraham et al. [53] | 2017 | 66.80 |
| Heinsfeld et al. [37] | 2018 | 70.00 |
| Eslami et al. [38] | 2019 | 70.30 |
| Wang et al. [54] | 2020 | 74.52 |
| Yang et al. [55] | 2020 | 75.27 |
| Tang et al. [39] | 2020 | 74.00 |
| Our Proposed Model | – | 87.87 |

The present study marked a significant performance improvement compared to existing studies. Despite that, some limitations need to be addressed. Only functional MRI data were utilized here for classification, whereas a combination of functional and structural MRI data has proven to achieve high prediction accuracy in [31,56]. Therefore, in future studies, other imaging modalities, such as structural MRI along with functional MRI data, may contain complementary information regarding ASD. However, implementation of the domain adaption technique [57] and encoding decoding technique [58] would also aid in prediction with more reliability and generalize well on unseen data obtained from different screening sites following different acquisition protocol. Other advanced neural network architectures, such as CNN, 3D based CNN model, etc., can also be utilized for prediction purposes and might prove to be fruitful. Furthermore, other options for implementing pipeline steps, such as usage of other available atlases, such as CC400, HO, Dosenbach, MSDL, etc., usage of first principal component-based time series extraction as in [59], non-correlation based functional connectivity matrix parametrization as in [60] and graph-based spectral method of vectorization as in [61], are aimed to be implemented in future studies.

## 4. Conclusions

In this paper, a deep learning approach using multisite resting-state fMRI was introduced to predict ASD. ASD detection is a challenging task since no standard modeling choice has yet been recognized, and the current practice is very much diverse. In this paper, preprocessed fMRI data were obtained from the CPAC pipeline. To extract mean BOLD signals from preprocessed data, brain atlases were used. A single brain atlas that can serve as a biomarker for the detection of ASD has not yet been discovered. Thus, four different standard and predefined atlases were used to extract ROIs. Connectivity matrices were prepared using tangent embedding and flattened to form a feature vector removing redundant information. This feature vector was provided as input to our proposed model. Hidden layer configuration of the model was also varied, and its impact on detection observed. After performing a wide array of experiments, it has been confirmed that the BASC atlas using 122 ROIs yields higher predictive power than AAL, CC200, or Power atlases and can be considered to be more reliable in ASD diagnosis. It achieved 88% accuracy, 90% sensitivity, 87% F1-score, and 96% area under receiver operating characeristic curve. This result transcends most of the performances of existing works indicating that it is a promising method for ASD diagnosis. The successful implementation of this method may be used for a wide range of applications, such as identifying neural activation patterns responsible for autism and performing visual evaluation of the functional characteristics of the autistic brain. By examining the contrast between the autistic and control brain, the underlying neural or biological basis of ASD can also be unveiled and established.

**Author Contributions:** Conceptualization, K.D.; Data curation, F.Z.S.; Formal analysis, F.Z.S.; Funding acquisition, P.K.D. and T.K.; Investigation, F.Z.S.; Methodology, F.Z.S.; Software, F.Z.S.; Supervision, K.D.; Validation, F.Z.S.; Visualization, F.Z.S.; Writing—original draft, F.Z.S.; Writing—review & editing, K.D., P.K.D., and T.K. All authors have read and agreed to the published version of the manuscript.

**Funding:** This research received no external funding.

**Institutional Review Board Statement:** Not applicable.

**Informed Consent Statement:** Not applicable.

**Data Availability Statement:** The authors have used the publicly available Autism Brain Imaging Data Exchange (ABIDE) dataset which is available in [41]. For experimentation and validation, the CPAC preprocessed mean time-series dataset has been obtained from the preprocessed connectome project (PCP) initiative in [44] for Craddock 200 (CC200) and Automated Anatomical Labeling (AAL) atlas. Mean time-series dataset for Bootstrap Analysis of Stable Clusters (BASC) and Power atlas has been collected from [50].

**Conflicts of Interest:** The authors declare no conflict of interest.

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
