# Peer review of "A Deep Learning Approach to Predict Autism Spectrum Disorder Using Multisite Resting-State fMRI"

_applsci, doi:10.3390/app11083636_

Round 1
Reviewer 1 Report
The paper entitled “A Deep Learning Approach to Predict Autism Spectrum Disorder using Multisite Resting State fMRI” reports about a deep neural network (DNN) model, which utilizes different brain atlases, to classify Autism Spectrum Disorder (ASD) patients from healthy controls.
The proposed approach outperforms the mean accuracy of the state-of-the-art methods. The paper is well written and interesting, but few concerns need to be addressed before publication.
MAJORS:
The only major concern is about the cross-validation method employed. Specifically, in section 3.1, it is stated that a 5-fold cross-validation is used, preserving the percentage of samples in each class to retain class balance. However, the 1D Feature Vector contains several samples from each subject, hence it is possible that samples from the same subject are in the training set, thus introducing a possible overfitting. Did the Authors consider this aspect? Please, comment and solve on this in the manuscript.
MINORS:
- Line 79, ROIs occurs for the first time here, but it is specified at line 133. Please extend the acronym at the first occurrence.
- Line 224, formula 2, please check if +1 needs to be subscripted
- Table 2,3, 4 and 5: it could be interesting to provide not only the mean value of the metrics describing the performance of the models, but also the standard deviation. Also in figure 11, it could be worthy to add the standard deviation.
Reviewer 2 Report
The article is devoted to the important problem of diagnosing autism based on resting state fMRI data from ABIDE dataset using deep learning methods. A fairly complete overview of the results achieved in this problem is given.
Resting state fMRI data for each subject is considered in the article as a collection of the time series for each voxel. Based on this presentation, a fairly standard pipeline has been proposed to solve this problem:
Step 1. Brain is parceled into regions according to chosen atlas using an appropriate tool (all voxels are divided into groups where each group consists of the voxels from corresponding region). This implies that a collection consisting of all the time series is divided into groups of time series (each group consists of time series for voxels of the corresponding region).
Step 2. For all voxels in each region, single ‘regional’ time series is constructed (averaged time series is used in the article).
Step 3. Functional connectivity matrix that describes statistical dependencies between regional time series is constructed (correlation connectivity matrix is used in the article).
Step 4, The symmetrical correlation matrix is transformed in the vector consisting of upper triangular matrix’s values.
Step 5. Using ABIDE dataset, deep learning network classifier is constructed for diagnostic purpose (using appropriate tool based on concrete chosen architecture).
In essence, the article represents the results of the computational experiments in which only the used anatomical atlas was varied: 4 atlases were considered and among them the atlas with the best quality of the classifier built on it was chosen.
There are a large number of known options for implementing pipeline steps (for example, usage of the first principal component in Step 2, usage of other functional connectivity matrices (non-correlation one) in Step 3, other matrix vectorization methods (such as spectral of graph-based ones) in Step 4, etc.). Therefore, the results of the article are of a rather particularistic nature.
The reviewer has doubts about the reliability of the article's results, which must be eliminated without fail. The doubts consist in the following. ABIDE dataset consists of data obtained in different scientific organizations and consists of 17 sites (this fact is emphasized even in the title of the article in the term “Multisite”). Different MRI scanners and protocols can be used in various organizations for receiving the resting state fMRI data, and known phenomena called domain shift, or distributional shift, takes a place. Conventional machine-learning algorithms often adapt poorly to domain shifts, and specific techniques (such as domain adaptation) are used to neutralize this effect. However, this issue is not even discussed in the article. Therefore, the article should at least briefly consider this issue and explain why this phenomenon did not affect the reliability of the article's conclusions (if not).
Reviewer 3 Report
Major comments
The biggest problem of this manuscript is that there is no description of the details of the method presented in this paper. As mentioned in the INTRODUCTION part, there are tons of challenges to distinguish the autistic participants from the TD (typically developed) participants without any diagnosis by psychiatrists, based on the behavioral signature, brain structure, and functional images. I agree with the authors’ claim that the discrimination power in previous studies is very low, and then there is a rest to improve the ability, and that the present study showed the overwhelming results. However, there is no description why previous models could not show enough ability, what mechanisms were introduced in this study to solve the shortage in previous studies, and what mechanism is indispensable to show their outstanding performance. In addition, the authors challenged to use the resting-state data acquired in different institutes. It is well known that there is a huge variance across data sets acquired in different sites, because of differences of a scanner, parameters, the way of lying participants in the scanner, and so on. What mechanisms or algorithms in this paper subserves for overcoming the big variance? No description, then I have no idea. The authors have to add the details first. Without them, I could not judge what the authors did is appropriate for.
I really doubt that the resting-state network could really be used to distinguish the participants with autism from TD participants. For example, He et al. (2020, HBM) reported the nonreplication of resting-state functional connectivity differences between the autistic participants and TD participants. How do authors think about it? I know that He’s paper does not mean it is impossible to machine-learning or deep-learning based method could well distinguish them. There is a possibility. However, it is impossible to judge whether or not the authors’ method could overcome it, because there is no details of the proposed method in this paper. In addition, the authors have to well think about the ratio between the ASD and TD participants. As mentioned in the Introduction part, there is almost 1 ASD subject in 160 TD. Is it possible to well distinguish them without any false discoveries, i.e. Type-II error? Finally, is the discrimination based on the neural activation? It is well known that the participants with ASD shows many sudden body movements, and the body movements have large effects on the functional connectivity. Then, it is possible that the discrimination between two groups is due to the differences of body movements, rather than neural activation. How the authors think about this?
Minor comments
Atlas selection: Why did the authors selected these four atlases? There are too many atlases those were not used in this study, and it is possible that the usage of these atlases could show super good performance.
L88: SPM8. Add the information of SPM8 distributors.
L224 (eq.2): It may be wrong. Zi+1? In addition, there is no description about what the z is.
Round 2
Reviewer 3 Report
I really appreciate the authors for their effort to answer my question. However, as a whole, the answer written in the response-to-reviewer is not reflected in the text. For instance, the authors explained me the details of what they did in the paper in step-by-step manner, and I (almost) understand what they did. I think this step-by-step manner is really good way. Then, authors should add the explanation written in the Response-to-reviewer to the manuscript itself. In the present version, I could (almost) understand your procedure, but not potential readers. The authors’ purpose is not let me understood, but let readers understood.
L88: “SPM8 distributor software. “
This is not what I intended. By googling, you could find the information about the SPM distributor, “Wellcome Centre for Human Neuroimaging”. The authors should show respects to the researchers provided research resources.
